# An Insight into Animal Glutamate Receptors Homolog of *Arabidopsis thaliana* and Their Potential Applications—A Review

**DOI:** 10.3390/plants11192580

**Published:** 2022-09-30

**Authors:** Ruphi Naz, Andleeb Khan, Badrah S. Alghamdi, Ghulam Md Ashraf, Maimonah Alghanmi, Altaf Ahmad, Sheikh Shanawaz Bashir, Qazi Mohd Rizwanul Haq

**Affiliations:** 1Department of Biosciences, Jamia Millia Islamia, New Delhi 110025, India; 2Department of Pharmacology and Toxicology, College of Pharmacy, Jazan University, Jazan 45142, Saudi Arabia; 3Department of Physiology, Neuroscience Unit, Faculty of Medicine, King Abdulaziz University, Jeddah 21589, Saudi Arabia; 4Pre-Clinical Research Unit, King Fahd Medical Research Center, King Abdulaziz University, Jeddah 21589, Saudi Arabia; 5Department of Medical Laboratory Sciences, Faculty of Applied Medical Sciences, King Abdulaziz University, Jeddah 21589, Saudi Arabia; 6Vaccines and Immunotherapy Unit, King Fahd Medical Research Center, King Abdulaziz University, Jeddah 21589, Saudi Arabia; 7Department of Botany, Aligarh Muslim University, Aligarh 202002, India; 8Department of Botany, Jamia Hamdard University, New Delhi 110062, India

**Keywords:** fluorescence resonance energy transfer, glutamate receptors, ligand binding domain, signaling

## Abstract

Most excitatory impulses received by neurons are mediated by ionotropic glutamate receptors (iGluRs). These receptors are located at the apex and play an important role in memory, neuronal development, and synaptic plasticity. These receptors are ligand-dependent ion channels that allow a wide range of cations to pass through. Glutamate, a neurotransmitter, activates three central ionotropic receptors: N-methyl-D-aspartic acid (NMDA), -amino-3-hydroxy-5-methylisoxazole-4-propionate (AMPA), and kainic acid (KA). According to the available research, excessive glutamate release causes neuronal cell death and promotes neurodegenerative disorders. *Arabidopsis thaliana* contains 20 glutamate receptor genes (AtGluR) comparable to the human ionotropic glutamate (iGluRs) receptor. Many studies have proved that AtGL-rec genes are involved in a number of plant growth and physiological activities, such as in the germination of seeds, roots, abiotic and biotic stress, and cell signaling, which clarify the place of these genes in plant biology. In spite of these, the iGluRs, *Arabidopsis* glutamate receptors (AtGluR), is associated with the ligand binding activity, which confirms the evolutionary relationship between animal and plant glutamate receptors. Along with the above activities, the impact of mammalian agonists and antagonists on *Arabidopsis* suggests a correlation between plant and animal glutamate receptors. In addition, these glutamate receptors (plant/animal) are being utilized for the early detection of neurogenerative diseases using the fluorescence resonance energy transfer (FRET) approach. However, a number of scientific laboratories and institutes are consistently working on glutamate receptors with different aspects. Currently, we are also focusing on *Arabidopsis* glutamate receptors. The current review is focused on updating knowledge on AtGluR genes, their evolution, functions, and expression, and as well as in comparison with iGluRs. Furthermore, a high throughput approach based on FRET nanosensors developed for understanding neurotransmitter signaling in animals and plants via glutamate receptors has been discussed. The updated information will aid in the future comprehension of the complex molecular dynamics of glutamate receptors and the exploration of new facts in plant/animal biology.

## 1. Introduction

In plants, there is continuous research on the glutamate (glu) receptor ion channel activity, but stock is still limited. The *Arabidopsis thaliana* genome is the first extensively sequenced genome and is considered a model plant for its conserved gene processes across all eukaryotes [1]. Among the 600 genes that are assumed to be associated with transmembrane protein encoding, 20 are predicted to be homologous to animal ionotropic glutamate receptors (iGluRs) [2]. These are known as *Arabidopsis thaliana* glutamate receptor channels (AtGluRs). Previously, it was believed that plants, unlike the mammalian nervous system, did not have any stimulus recognition sites. However, with the assistance of the famous scientist Sir Acharya J C Bose [3], it was demonstrated that plants have a distinct manner for their sensitive neurological systems, and it was confirmed that plants feel pain and affection, not precisely as in an animal system, but they respond to external stimuli [4] and to different signals such as light, cold, hot, and wounding.

Ionotropic glutamate receptors (iGluRs) are generally conserved genetic elements in vertebrates and have been identified as potentially playing a function in cell signaling and rapid excitatory neuronal transmission in mammalian central nervous systems [5]. The involvement of iGluRs in the brain’s learning and retention power, anxiety, and chronic pain, as well as for the indication and detection of Alzheimer’s disease (neurodegenerative disease) is critical [6,7]. The iGluRs family is made up of non-NMDA (AMPA and Kainate), NMDA, and Delta or orphan receptors (Figure 1). These receptors are nonselective tetrameric cation channels that bind to ligands. When the ligand (glutamate) attaches to the ligand-binding pocket of these receptors, cations such as Na^+^, K^+^, or Ca^2+^ enter the cell [8]. However, numerous existing evidence of plant glutamate receptors being comparable to animal glutamate receptors has piqued the interest of researchers for a long time, aiming to uncover fascinating facts about glutamate receptors. A number of researchers are engaged in the study of glutamate receptors exploiting the model plant *Arabidopsis thaliana* (L). These AtGluRs genes can be a guide towards understanding the functional features of the GluR family’s genes. Research has shown that the model plant *A. thaliana* possesses 20 potential glutamate receptor gene families, which are involved in signaling [9] and have a tight relationship with the glutamate receptor of vertebrates, indicating that they must have originated from a common ancestor [10,11,12,13]. Moreover, Chiu et al. [12] determined, through an initial comparative phylogenetic analysis of plant GluRs and animal iGluRs, that signaling through amino acids was supposed to have existed earlier during the divergence of animals and plants. Plant GluRs have a pivotal role in the metabolism of nitrogen and carbon [14], growth in the pollen tube [15,16], gravitropism [17], defense signaling [18,19], and in wound signaling [20].

In addition, plant glutamate channels are not just restricted to the plasma membrane; they may also be found in the mitochondria, vacuolar membranes, and chloroplasts [21]. Furthermore, many researchers are constantly striving to use additional elements such as gene tagging, protein−protein interactions, and intact dye-based (green fluorescent protein) and fluorescence-based high-throughput approaches in order to accomplish the deep core of glutamate receptors in *Arabidopsis*. The present review first emphasizes the updated information on *Arabidopsis* glutamate receptor genes, their co-relation with animal ionotropic glutamate receptors, and their expression patterns. Following this, we presented the construction of genetically encoded fluorescent glutamate nanosensors based on fluorescence resonance energy transfer (FRET). This information will help future engineering and applications by using high-throughput techniques for readers to examine *Arabidopsis* glutamate receptors.

## 2. Plant GluRs: Evolutionary Authentication

The 20 AtGluRs were classified into three evolutionary clades [12]. Data collected by sourcing the bacterial periplasmic protein for the amino acid binding of 20 *Arabidopsis* GluR-genes, whole rat iGluRs, and other two iGluRs of prokaryotes such as *Synechocystis* GLRO and *Anabaena* iGluRs showed that signaling behavior existed prior to the divergence of animals and plants, indicating that they evolved from a common ancestor [12]. The sequence of amino acids and various channels such as potassium channels, GABA, and acetylcholine receptors demonstrated that plant GluRs are identical to animal ionotropic glutamate receptors. Using a phylogenetic tree, it has been confirmed that plant and animal GluRs emerged from a common ancestor, as shown in Figure 2 [12,22].

Furthermore, Turano et al. and Nagata [23,24] proposed that *Arabidopsis* glutamate receptors, such as calcium sensor and glutamate/4-aminobutanoic acid (GABA) protein receptors, occupy a long N terminal to establish the evolutionary relationship between glutamate receptors of different species [22,25]. According to Price et al. [26], from an evolutionary standpoint, plant GluRs diverged from eukaryotic GluRs in an ancient time zone. Thus, the combined literature suggest that plant homolog GluRs should be placed in a different evolutionary position than metazoan glutamate receptors.

## 3. Structure of *Arabidopsis* Glutamate Receptors

The structure of AtGluRs is made up of 800–960 amino acids containing a molecular weight of ~100 kDa per receptor and six conserved domains, similar to the animal glutamate receptors [10]. These structures have S1 and S2 sites that allow for ligand and M1 to M4 transmembrane domains with a pore region (P), which aids in ion passage management [9]. S1 and S2 domains, such as iGluRs, undergo conformational changes when ligands bind to it [27]. The N-terminal domain of both plant and animal resides outside of the plasma membrane. As a result, it is assumed that these receptors follow a similar route in both plants and animals (Figure 3). The research of Ulbrich and Isacoff [28] revealed that N-methyl-D-aspartate (NMDA) receptors are functionally active when N-methyl-D-aspartate-selective glutamate receptor (NR 1, 2 and 3) subunits of NMDA are put together in a particular homo- or hetero-tetrameric combination. Receptor complexes are assembled inside the endoplasmic reticulum to interact with the N-terminal domains of these receptors [29,30]. In addition, similar to animal iGluRs, plant receptors (AtGL-receptor 2.9, 3.2, and 3.4) create interactions through their N-terminal domain with other glutamate receptors [31]. Several studies have shown that glutamate receptors exist in homo-/hetero-imeric form. The experiment of Stephens et al. [32] suggested that because of the heteromeric channels, channels “A” are triggered by the application of Glu, channels “B” by different (Ala, Cys, Gly, and Glu) amino acids, and channels “C” by all six ligands (Ala, Asn, Cys, Glu, Gly, and Ser). Based on the nature of the channel activation, they confirmed that at least one subunit is essential to the function of channels, as channels “A”, “B”, and “C” hold one subunit of AtGluR-3.3, AtGluR-3.4, and both AtGluR-3.3/3.4, respectively. Thus, all channels are triggered by Glu, and perhaps AtGluR--3.3 is an essential subunit (Figure 4).

Using FRET approaches, Vincil et al. [33] showed homo-/hetero-multimerization in the AtGluRs (3.2 and 3.4) through their expression in tobacco leaves. Other approaches, such as the yeast two-hybrid (Y2H) system, confirmed the interaction of the homo-/hetero-meric production of AtGluRs on the plasma membrane [34].

## 4. Similarity between Plant and Animal Ligands and the Ligand Binding Domains

The study of Lam et al. [10] proposed that, along with conserved membrane motifs (M1 to M4), glutamate receptors are also comprised of two separate ligand-binding domains found in the outer region of the membrane. S1 and S2 are two putative substrate-binding regions of a domain and participate in ligand binding directly [10,31]. Further research has demonstrated that the 3D structure of S1 and S2 shows a clamshell-like structure [35,36,37,38,39]. This site consists of lysine/arginine/ornithine-binding protein (LAOBP), which is homologous to the periplasmic binding protein II superfamily [40]. The literature has revealed that the ligand-binding domain shows conformational changes when ligand molecules bind to it, i.e., they undergo a “venus-flytrap”-like movement. Thus, glutamate receptors are very good, specifically for agonists [38,41,42,43].

Based on the evolutionary history, transmembrane domains and ligand binding sites evolved prior to the divarication of plants and animals. Different scientist groups [12,24] have worked on glutamate receptor homology and concluded that there is about ~50–60% similarity between animal and plant glutamate receptors, with the M2 domain showing a lower homology than the M3 domain, which shows 61% [22,44]. The S1 and S2 ligand binding domains consist of conserved residues [22], which was supported by Zuo et al. [45] through the mutation in the M3 region in Lurcher mice. Four of the six transmembrane domains shared by the three clades were substantially similar. In addition, Chiu et al. [12] reported that among the three clades of 20 AtGluRs, domains M1, M2, and M3 appear highly conserved, whereas S1 and S2 show differences. These results show that these receptors bear the capacity to bind their own ligands, including amino acids or other molecules. Other than these conserved domains, a unique domain identical to the G protein-coupled receptor (GPCR) has been identified in almost all plant glutamate receptors. A number of substrates have been identified to bind to animal glutamate receptors, such as aspartate (Asp), homocysteine (Cys), glutamate (Glu), lysine (Lys), d-serine (Ser), arginine (Arg), and glycine (Gly) etc [46,47,48].

Although AtGluRs are not fully characterized [19], many studies have revealed that the binding of glutamate and other amino acids to the plant glutamate receptors has the potential to increase membrane depolarization, through which the conductance of ions could be able to suggest the presence of amino acid gated calcium ion channels in plants [33,49,50,51]. Furthermore, Vincill et al. [33] suggested that plant glutamate receptors are capable of forming amino acid gated channels, which was proven by aspargine (Asn), Gly, and Ser binding with AtGluR-3.4 in expressing mammalian cells. Many research groups are actively working to reveal that the plant ligand-binding domain (LBD) is similar to the mammalian LBD. Tapken et al. [52] demonstrated that in AtGluR-1.4, a small change in amino acids T501, R506 1, and D499 suppressed the ligand-gated channel activity when expressed in the *Xenopus oocytes*, which clearly shows that these are the conserved amino acids in AtGluRs. In addition, it seems that as the animal iGluRs, the α- amino and β carboxyl primary amino acid groups are essential for bind in the plant GluRs [31,52]. The study of Standley and Baudry [53] concluded that glycosylations of the amino acid chain, which could be the cause of alterations in the receptors binding phenomenon, are likely similar in function in the plant and animal glutamate receptors.

## 5. The Activity of Glutamate Receptors in Plants

The iGluRs antagonist DNQX (6,7-dinitroquinoxaline-2,3-dione) was used to detect the glutamate activity in plants for the first time in the form of hypocotyl development under light conditions [10]. According to Dubos et al. [50], glutamate and glycine, similar to the animal-GluR, successfully controlled hypocotyl development; however, the animal-GluRs inhibitor DNQX decreased hypocotyl elongation. Michard et al. [15] employed 6-cyano-7-nitroquinoxaline-2,3-dione (CNQX), DNQX, and 2-amino-5-phosphonovalerate (AP5) antagonists on pollen tubes (pt) to validate the glutamate receptors in plants, and reported a considerable decrease in tobacco plant growth. The agonist D-Serine, on the other hand, was capable of promoting pt proliferation in tobacco. According to Meyerhoff et al. [13], glutamate-dependent calcium signals were lowered in *A. thaliana* mesophyllous cells owing to antagonism 5,7-Dinitro-1,4-dihydro-2,3-quinoxalinedione (MNQX, DNQX, and CNQX).

Based on biochemical and electrophysiological experiments, Teardo et al. [54] provided information for GluRs in the chloroplast in spinach. Li et al. [55] revealed that GluRs interacted with a potential glutamate ligand in *A. thaliana.* The role of GluR was confirmed in *A. thaliana* by utilizing BMAA[S(+)-beta-methyl-alpha, beta-diaminopropionic acid] on *Arabidopsis* seedlings and hypocotyl elongation and cotyledon opening inhibition was found compared with the control plant [56]. Several studies have also shed light on the specificity of amino acids for *Arabidopsis* GluRs. AtGluR-1.4 was performed with seven amino acids (tryptophan, methionine, phenylalanine, leucine, tyrosine, asparagine, and threonine), whereas AtGluR-3.4 was gated by three (Asn, Ser, and Gly) [33,52]. The study by Tapken et al. [52] was able to conclude that AtGluR-1.4 functioned as a nonselective cation channel. In contrast, the evidence of Vincill et al. [33] found that AtGluR-3.4 was highly sensitive to Ca^2+^ permeability. In this way, these results showed that, similar to animal GLuR, also in the planta, glutamate and amino acids fully participated in membrane depolarization and the influx of Ca^2+^ [26].

## 6. AtGluRs in Response to Agonists and Antagonists

A review of the literature indicated that animal glutamate receptor agonists and antagonists had almost the same effect on plants as they do on animals. Previously, glutamate was thought to be a particular agonist, but the study of Dubos et al. [50] uncovered that, in addition to glutamate, glycine is the major component of almost all AtGluRs. According to Dubos et al. [50], in almost all AtGluRs, highly conserved threonine (Thr) for glutamate binding may be substituted with phenylalanine (Phe). Kang and Turano [14] studied AtGluRs-1.1 and proposed that antisense AtGluRs-1.1 was not hypersensitive to K^+^/Na^+^ when compared with the wild type, but high Ca^2+^ concentrations inhibited root development. They [14] demonstrated that glutamate is not a real ligand, but BMAA is a legitimate AtGluRs-1.1 ligand engaged in carbon and nitrogen sensing. Furthermore, Walch-Liu et al. [57] demonstrated that exogenous glutamate caused a significant shift in *A. thaliana* root growth.

The effect of the agonist (BMAA) and antagonist (DNQX) revealed that the presence of the antagonist inhibited the hypocotyl elongation [10], whereas the agonist promoted hypocotyl elongation [10,56]. Other than Glu and Gly, a variety of amino acids (serine (Ser), methionine (Met), asparagine (Asn), alanine (Ala), glutathione, and cysteine (Cys)) respond to AtGluRs [32,33,49,51,52,55]. In *Arabidopsis*, the agonist D-Ser was the most potent at increasing the Ca^2+^ concentration in pollen tubes, but when the antagonist CNQX was applied, pollen tube deformation was observed [15].

However, the impact of the agonist/antagonist on AtGluR-1.3 is still not clear. A significant fraction of methionine, tryptophan, tyrosine, leucine, and phenylalanine in *Arabidopsis* seedlings was considered by AtGluR-1.4 [52]. This amino-acid-induced depolarization in *Arabidopsis* leaf cells is considered a potent agonist of AtGluR-1.4. This observation provides the possibility that methionine is not only considered a nitrogen molecule, but also serves as a signaling molecule [52]. DNQS, CNQS, and MK-801(non-competitive NMDA receptor open channel blocker) have been reported as antagonists for AtGluR-1.4, which are similar to the animal antagonist [10,13,14,15,52,54,55,58,59,60] (Table 1). Electrophysiological evidence has shown that Met works as an agonist to supply Ca^2+^ in AtGluR-3.1/3.5 [61]. As with the other AtGluRs where different amino acids serve as agonists [15,19,52,61], a similar effect of different amino acids was observed for AtGluR-3.2, and D-serine was found to be the most relevant agonist [62].

In addition, Glu, GABA, aspartate, and malate worked as agonists on AtGluR-3.4. To see the effect of antagonists [13], different animal antagonists, DNQX, CNQX, and MNQX, were tested and it was observed that these antagonists were susceptible to glutamate-dependent Ca^2+^ transient [13]. Other than amino acids, kanamycin and other polyamines are also considered to be glutamate receptor agonists [63]. The study of Tapken and Hollmann [64] showed that AtGluR-1.1 pore domains allow for Na^+^, Ca^2+^, and K^+^ ion permeability. The flow of these match the glutamate triggered Ca^2+^ influxes. As b-methylamino-L-alanine (BMAA) works as an agonist of animal iGL-rec, the application of BMAA has some contradictions as it shows a similar activity to an agonist [56] or antagonist [65,66]. Further evidence comes from the research of Singh and Chang [67], that GluR exhibited an antagonist (CNQX/DNQX) behavior and leads to alterations in the root.

In contrast, the agonist (L-glutamate) helps recover root growth. Using different antagonists (DNQX, AP-5, and MK-801), Walch- Liu and Forde [65] demonstrated that no effect was observed on plant growth as a reference of antagonism, which contradicted Dubos et al. [50], who demonstrated that DNQX could perhaps attach to binding sites of the ligand and mimic Ap-5 mimic as L-glutamate, bind on that site, and halt growth. However, a number of research groups have observed the effects of DNQX, CNQS, memantine, and MK-801 on AtGluR [10,13,14,15,52,54,55,58,59,60].

## 7. Function, Expression and Applications of AtGluR Genes

Expression studies have shown that plant glutamate receptor genes are successfully expressed in plant organs such as the leaves, roots, and reproductive bodies [12,13,68]. Clade I and III genes were thought to be expressed in almost all plant parts, whereas Clade II genes were not; however, later it was observed that these genes were identified in complete plants [12,69]. The expression and their functional applications have been described based on the observations of several researchers.

### 7.1. Clade I

#### 7.1.1. AtGluR-1.1

The AtGluR-1.1 gene has been reported to play a role in potential activity by regulating and signaling abscisic acid (ABA) biosynthesis in *A. thaliana* [58]. AtGluR-1.1 also regulates C, N, and water balance, which are essential for plant growth. The GUS expression was first observed in stipules and the collette region of 7-day-old *Arabidopsis* plants. Later, its expression was found in the leaf margins and in the cells of the lateral roots (Figure 5). A low expression of GluR-1.1 was also detected in the flowers, siliques, and reproductive organs [12].

#### 7.1.2. AtGluR-1.2 and 1.3

Zheng et al. [70] demonstrated that GluR-1.2 and -1.3 regulate cold tolerance in *Arabidopsis* owing to cold stress by stimulating endogenous jasmonate synthesis, and these genes also play a crucial role in the downstream CBF/DREB1pathway during cold stress [70,71] (Figure 6). A fluorescent tag was ligated with GluR-1.2 and -1.3 to detect its expression, and an increased fluorescence was observed in the plasma membrane of *Nicotiana benthamiana* [70].

#### 7.1.3. AtGluR-1.4

The expression of GFP-tagged AtGluR-1.4 in the plasma membrane of wild-type *Arabidopsis* plants was studied. StREM1.3, a plasma membrane marker tagged with red fluorescent protein (RFP), was also employed for co-localization [52]. However, Roy et al. [72] detected a varying expression of AtGluR-1.4 in different cells of the same plant, as well as in distinct plants. In addition, Ca^2+^ depolarization in leaf cells is perhaps caused by amino acid signaling [52] (Figure 7).

### 7.2. Clade II

#### 7.2.1. AtGluR-2.1

On the basis of the phylogenetic and expression analysis of the glutamate receptor, Chiu et al. [12] reported that expression of a clade-II gene (GluR-2.1) was observed in the shoot of a 5-day-old plant that showed similarity with GluR-1.1. In addition, GluR-2.1 was also detected in the radical of emerging seedling, and it was expended in all cells of the root, except in the root apex of 3-day-old seedling. However, a slight expression was detected in the reproductive organs and no such expression was shown in the siliques or flowers [12]. Roy et al. [72] found that AtGluR-2.1 was expressed in 4-week-old leaf tissue. The pretreatment of glutamate in plants enhanced the upregulation of the gene expression and the accumulation of amino acids [73] (Figure 8).

#### 7.2.2. AtGluR-2.2

The expression of AtGluR-2.2 was reported in the root, whereas it was lost in reproductive organs and the leaves [12]. A similar observation was reported by Roy et at. [72], where AtGluR-2.2 was expressed in the root but was absent in leaf, stem, and petiole part of the plant.

#### 7.2.3. AtGluR-2.3

The expression of AtGluR-2.3 was quite different, as it was restricted in 8-week-old plants; after that, it was seen in the root tissue [12]. AtGluR-2.3 was not expressed in the flowers and silique, while a deficient expression was detected in the leaves of the developing plant [72].

#### 7.2.4. AtGluR-2.4

In addition, AtGluR-2.4 was expressed in the root and silique, but the expression was completely stopped in stem, petioles, and leaf tissues [12,68,72].

#### 7.2.5. AtGluR-2.5

Based on the mRNA expression analysis, Chiu et al. [12] reported that AtGluR-2.5 was expressed in the whole plant tissues.

#### 7.2.6. AtGluR-2.6

It was expressed in the root [12], and was absent in the leaf, stem, and petioles [72].

#### 7.2.7. AtGluR-2.7

All of the plant parts showed a higher expression of AtGluR-2.7 instead of the flowers [12,74].

#### 7.2.8. AtGluR-2.8

Along with the root and shoot, a good expression was also observed in the leaf mesophyll cells around the vascular bundles with a GUS straining and expression became higher in the leaves during the senescence stage [68,74] (Figure 8).

#### 7.2.9. AtGluR-2.9

Similar to the other Clade II receptors, the AtGluR-2.9 gene was also expressed in the root [68]; however, Roy et al. [72] observed that gene 2.9 either showed a lower expression in the leaves or was completely stopped (Figure 8).

### 7.3. Clade III

#### 7.3.1. AtGluR-3.1

In order to comprehend the expression and function of AtGluR-3.1, Kong et al. [61] investigated AtGluR-3.1 in depth and fused a green fluorescent protein (GFP) tag with GluR-3.1 and 3.5, determining that these proteins were situated in the plasma membrane and were expressed in the guard cells. Furthermore, the study found that these receptors were expressed in the seedling cells, other than the guard cells. Cho et al. [75], on the other hand, suggested that an increased expression of AtGluR-3.1 was seen in the guard cells, followed by the mesophyll cells. Furthermore, the deregulated expression of AtGluR-3.1 altered the stomatal closure, without impairing cytosolic Ca^2+^.

#### 7.3.2. AtGluR-3.2

The expression of AtGluR-3.2 was confirmed in all parts of the plant and was highly expressed in the root cells [33]. The exogenous supply of Ca^2+^ demonstrated the function of AtGluR-3.2 to overcome the Ca^2+^ deprived condition, which was induced through overexpressed AtGluR-3.2 in transgenic plants [44].

#### 7.3.3. AtGluR-3.3

According to Li et al. [55], supplementation of exogenous GSH activates the AtGluR-3.3 and participates in the early transcriptional process of the leaves (Table 2). However, the genetic process underlying GSH and the independent expression of the gene is not known. Furthermore, root gravitropism in AtGluR-3.3 mutant lines is likely to diminish amino acid regulated Ca^2+^ signaling [17].

#### 7.3.4. AtGluR-3.4

Among the other AtGluRs, AtGluR-3.4 is strongly expressed in guard cells, vascular bundles, roots, and mesophyll and stem. AtGluR-3.4 responds under abiotic stimuli such as cold and touch in a Ca^2+^ dependent manner. Along with the different expression, a weak expression was also observed in the cortex, root epidermis, and hairs. However, the activity of the cold expression was halted by adding lanthanum (Ca^2+^ channel blocker). Moreover, a mutation in AtGluR-3.4 enhanced sensitivity towards ABA, which caused an impact on seed germination [76].

#### 7.3.5. AtGluR-3.5

The expression of AtGluR-3.5 predominantly appeared in *Arabidopsis* germinating seeds, where this receptor enhances the Ca^2+^ concentration. No significant expression was seen in the dry and mature seeds. However, repression in AtGluR-3.5 impacted Ca^2+^ signaling and was more sensitive to ABA, which exhibited a delay in seed germination. In contrast, a higher expression of AtGluR-3.5 was less sensitive to ABA and led to early germination. The site of expression of AtGluR-3.5 was confirmed by creating a transgenic plant with the GUS gene and it was identified that AtGluR-3.5 was expressed in the cotyledons of the germinating embryo [77].

#### 7.3.6. AtGluR-3.6

The study of AtGluR-3.6 demonstrated that the expression of AtGluR-3.6 is based in the developmental stages of the roots. Singh et al. [78] reported a higher expression level in the early stages of root tissues than the mature root tissues. The overexpression of AtGluR-3.6 promoted root growth. Furthermore, KRP4 (kip related protein) played a crucial role in maintaining AtGluR-3.1-related root meristem [78]. Cell cycle regulation also influenced the formation of the main and lateral roots [78,79]. Furthermore, it has been suggested that AtGluR-rec-3.6 may play a role in leaf wound signaling via the jasmonate pathway [20]. Other research indicated that AtGluR-3.6/3.3 played a vital role to induce Ca^2+^ elevation on the aphid feeding sites [80].

#### 7.3.7. AtGluR-3.7

Considerably, research on AtGluR-3.7 has found that this receptor is likely expressed in every plant cell and serves an important function as an ion transporter in *Arabidopsis* [72]. Further research has demonstrated that AtGluR-3.7 responds to salt stress in *Arabidopsis* via calcium signaling and is involved in seed germination and regulation [81,82]. The overexpression of AtGluR-3.7 provoked a marked increase in root growth.

### 7.4. AtGluRs Gene Respond to Environmental Stress

#### 7.4.1. Stress Due to Salt

Salt stress alters plant behavior and adaption mechanisms, which are usually mediated by the Ca^2+^ signaling network [13,83,84,85,86]. The study of Cheng et al. [87] demonstrated that AtGluR-3.7 played a significant role during seed germination under salt stress conditions. Furthermore, Cheng et al. [88] confirmed that salt stress treatment on *Arabidopsis* wild and mutant plants revealed that AtGluR-(3.4-1/2) of mutants were more sensitive in seed germination when compared with the wild type. An expansion study showed NaCl caused a rise in Ca^2+^ wave in the wild variety of plants, which was blocked by inserting an animal glutamate receptor antagonist (DNQX). However, an increase in Ca^2+^, triggered by NaCl, was impaired in mutated AtGluRs (3.4-1/2). In addition, more Na^+^ was accumulated in the mutant during salt stress than in the wild-type seeds. Overexpressed AtGluR-3.4 was highly resistant to abscisic acid [88]. Furthermore, Wang et al. [81] reported in their study that AtGluR-3.7, in conjunction with other proteins (14-3-3), plays a potential role in *Arabidopsis* under salt stress by influencing the Ca^2+^ signaling pathway. As a result of these findings, seed germination under salt conditions was regulated by Ca^2+^ inflow, which was modulated by AtGluRs-3.4/3.7. Therefore, the Clade III gene, GluR-3.7, plays a significant role in salt stress in *A. thaliana* [89].

#### 7.4.2. Stress Due to Cold

In addition to salt stress, cold stress is also another factor that harms plants. In 1999, Thomashow [71] reported that temperate region originating plants such as spinach and *Arabidopsis* showed cold tolerance upon exposure to low temperatures. This phenomenon is known as cold acclimation. In addition to the plasma membrane, chloroplast also plays a sensing role at low temperatures. Earlier, Meyerhoff et al. [13] suggested that because of the osmotic stress in *Arabidopsis,* AtGluR-3.4 was expressed in an ABA-independent fashion and provides strength in fast signaling via Ca^2+^ ion.

In transgenic plants, cold and Glu were able to induce Ca^2+^ ion channels, which was blocked by the antagonist (DNQX/CNQX). The study of Hu et al. [90] indicated that during cold stress, phytohormones altered their activity and response towards plants. Further research revealed that the membrane protein sensed cold stress and perhaps activated Ca^2+^ signal transduction pathways. These signals may lead to activating the downstream transcriptional regulatory cascade and allow them to cope and survive in cold stress [42,91,92,93,94]. Thus, exposure to cold temperature plants induces tolerance in cold conditions, as well as a massive alteration in gene expression and Ca^2+^ signal transduction pathways. In addition, Zheng et al. [70] unveiled that AtGluRs-1.2 and -1.3 mutants performed a positive role towards cold stress in *Arabidopsis*, whereas with the influence of the plant hormone jasmonate, the activity of AtGluRs-1.2/-1.3 mutants to cold stress was reduced. Furthermore, under a freezing environment, the expression of these mutants was lower in the C-repeat binding factor/DRE binding factor 1 (CBF/DREB1) transcriptional regulatory pathway compared with the wild-type plant. In brief, Zheng et al. [70] suggested that AtGluRs-1.2/1.3 could increase cold tolerance by trigging endogenous jasmonate accumulation (Figure 9).

#### 7.4.3. Stress Due to Drought

Drought stress is a severe environmental barrier to plant productivity. Drought stress impacts different physiological processes and reduces crop yield [95,96]. Water scarcity has been observed at a morphological to molecular level in plants. Drought stress severely impacts on the photosynthetic abilities of plants, which imbalances metabolic activities. When faced with stress, plants modify their strategies to allow them to be capable of resisting and adapting to drought conditions [95,96]. After many debates on the closure of the stomata, its impact on drought stress or metabolic impairment was found. It has been proven that stomatal closure is one of the main factors for controlling water deficits as well as gas exchange during stress, which is regulated by the co-ordination of different signaling molecules, including Ca^2+^, H_2_S, reactive oxygen species, ABA, and nitric oxide NO [73,95,96]. The study of Yoshida et al. [96] demonstrated that glutamate (Glu), a signaling molecule that increases Ca^2+^ in guard cells, is responsible for closing the stomata and preventing light from opening up the stomata in *Arabidopsis,* as well as in *Vicia faba*. Furthermore, to discover the complete relevance of glutamate receptors with stomatal closure due to glutamate, different Ca^2+^ chelators such as EGTA (extracellular Ca^2+^ chelator) and BAPTA-AM (intracellular chelator) were used, and it was found that these chelators prevent the stomata from being closed, which is induced by glutamate, and show that the influx of Ca^2+^ into the cytosol is required. They also observed that Glu-dependent Ca^2+^ flow triggers Ca^2+^ dependent protein kinase and endorse the SLAC activity to close the stomata [96]. Other than *Arabidopsis*, Glu treatment in *Brassica* responds by triggering Ca^2+^ signaling, which increases proline accumulation due to drought stress and manages drought tolerance [97]. In addition, Philippe et al. [98] used animal glutamate receptor antagonist-AP-5, DNQX, in *Medicago truncatula* and reported that these antagonists were responsible for reducing NO accumulation, which showed the least effect on stomata closure. Furthermore, two glutamate receptors obtained from rice (GLR1 and GLR2) exhibited drought tolerance in *Arabidopsis* [99].

### 7.5. Role of AtGluRs in Plant Defense Signaling

Plants have defended themselves by evolving various defense tactics against the opponents, which take the shape of insects, pests, and other herbivores. Plants use both chemical and mechanical means to detect herbivores. Many recent studies have focused on monitoring early defense signaling to protect them from herbivore attacks in the form of fluorescent-based genetically encoded nanosensors to detect in real time signaling and modulate the ion channels that are commonly associated with plant−herbivore interactions [100]. Furthermore, a number of studies have shown that Glu receptors protect plants from insect injury. Because of insect activity on plants, Glu receptors increase Ca^2+^ flow, causing defensive signals to change in order for the plants to survive [49,60].

In an experiment when *A. thaliana* was wounded by *S. littoralis,* it was found that AtGluR-3.3 induced Ca^2+^ ion flow to reduce the surface potential caused by the larvae and showed a potential role in altering the defense signaling in plants [20]. Glu receptors are involved in plant defense signaling as they are activated by the wound [20,79]. In addition, recent research suggested that GluRs-3.3/3.6 modulates Ca^2+^ ion flux to defend the plants against wound-creating organisms [100,101]. Initially, the research of Kang et al. [102] showed that in transgenic *Arabidopsis,* overexpressed radish GluR intermingles with the increased expression of some defense-linked genes and accelerates the resistance against fungal pathogens. Afterwards, pharmacological studies showed that iGluRs antagonists were found to take part in the immune response in seedlings of *A. thaliana* [82] and in the suspension culture of tobacco [60]. In AtGluR-3.3 mutants, because of the attack of a bacterial pathogen (Pseudomonas syringae), susceptibility increased with regard to immune response, which exhibited the inadequacy in the activity of defense-related gene expression against infection [55]. Although almost all AtGluR genes take part in the defense mechanism, AtGluR-3.3 was found to be more responsible, as in *P syringae,* activation of the defense gene in response to enhancing immunity against infection depends on AtGluR-3.3 [51]. A study suggested that the glutamate receptors associated with Ca^2+^ influx are essential for initiating downstream signaling processes towards the plant defense. Further involvement of AtGluR-3.3 is important for resistance infected by *Hyaloperonospora arabidopsis (oomycete pathogen)* [18]. It has been found that the Clade III gene fully participates in the defensive mechanism against mechanical wounding caused by feeding insects and pests [20,103].

### 7.6. Developed FRET Based Glutamate Receptors Nanosensors

Modern biosensors based on FRET provide a potential tool. FRET has become an advanced phenomenon for monitoring the interactions, such as molecule to molecule or protein to protein, or conformational changes (Figure 10). The biggest advantage is that the distances are obviously shorter than the diffraction limits of other microscopies. Fluorescent proteins (FP) are used as an essential component in the creation of biosensors. This method is based on the excitation of the donor fluorophore, followed by the non-radiative transfer of the excited energy to the related other fluorophore protein, known as the acceptor molecule [104,105]. Various FRET-based nanosensors have been constructed, but we concentrated on FRET-based approaches to glutamate receptors. Glutamate receptors are not only for animal neurons and glial cells [106]. These receptors exist in the entire world, as well as eukaryotes and prokaryotes [19,107,108]. Okumoto et al. [109] were the first to construct a glutamate ratiometric sensor by sandwiching a truncated binding protein (gltl) of glutamate/aspartate between a fluorophore protein pair (enhanced cyan fluorescent protein (ECFP), mVenus). Hires et al. [110] introduced EYFP (enhanced yellow fluorescent protein) in lieu of m-Venus and conducted systematic monitoring of the glutamate affinities. Next, Marvin et al. [111,112] proposed that single-wavelength fluorescence receptors (FR) could detect neurotransmitters with a great spatial and temporal resolution. They created a number of glutamate iGluSnFR sensor versions employing blue, green, cyan, and yellow color emissions and circular permutated cp-GFP, with activities ranging from sub-micromolecular to millimolar.

The plant glutamate receptor protein family emerged as a binding cassette for neurotransmitters with the introduction of animal glutamate nanosensors. In a similar way to how glutamate acts in neurons, it also aids in the flow of impulses in plants [79,113]. Plants use the fluorescent indicator protein for glutamate (FLIPE) [113] (Figure 11). According to Forde and Roberts [19], glutamate receptor proteins have been found to bind GABA and amino acids such as glutamate/glycine. The high-affinity GABA transporter (AtGAT1) in *A. thaliana* may be required for GABA sensor function, as found by Meyer et al. [114]. To change the spectral range of the glutamate sensor, Wu et al. [115] created circular permutated (cp) R-iGluSnFR1 using red Glu sensing FR and noncircular permutated (ncp) Rncp-iGluSnFR1.

## 8. Conclusions

Neurotransmitters govern the ionotropic receptor channels (iGluRs) activity. iGluRs homologous in *Arabidopsis* have been significantly associated with plant physiological activity, mainly root development, protein and cell signaling, ion transport, and other metabolic pathways. Many studies have shown that the evolution of these AtGluRs are linked to that of animal glutamate receptors. Although valuable research has been carried out to consider different aspects of glutamate receptors. Further research is being utilized to discover the hide tract between animal and plant glutamate receptors. However, there are many challenges and pitfalls, but some solutions also exist. Recently, genetically encoded FRET nanosensors have been explored for an understanding of the plant biology. Still, there is a large window for further improvements related to animal and plant glutamate receptors. In this review, we tried to gather all of the updated information to consider every aspect of the *Arabidopsis* glutamate receptor gene. This review will contribute further knowledge to the readers.

## Figures and Tables

**Figure 1 plants-11-02580-f001:**
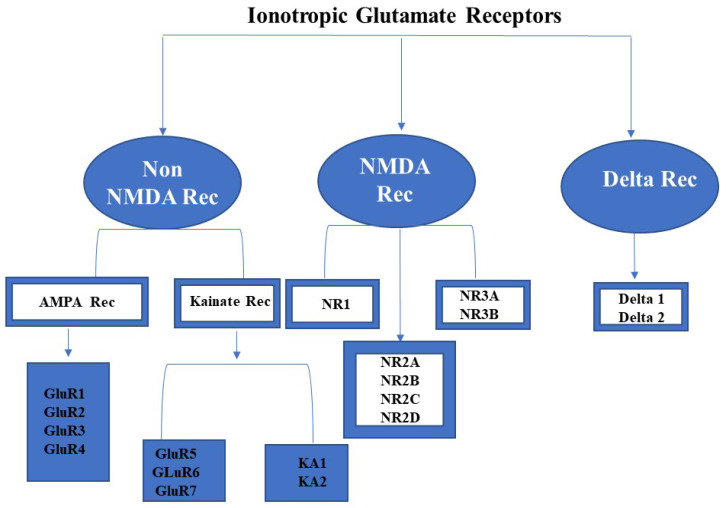
Different types of ionotropic glutamate receptors.

**Figure 2 plants-11-02580-f002:**
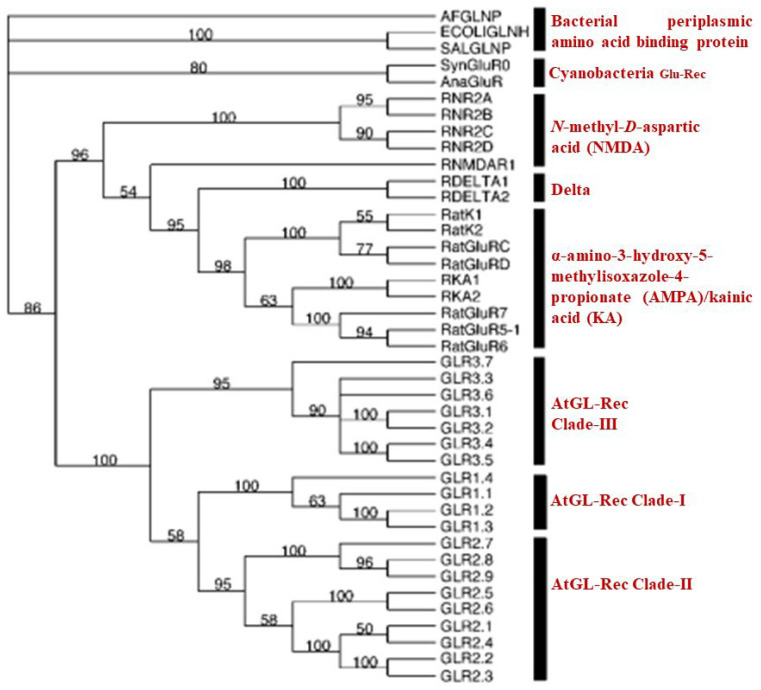
Evolutionary tree of AtGluR- proposed by Chiu et al., 2002 [12].

**Figure 3 plants-11-02580-f003:**
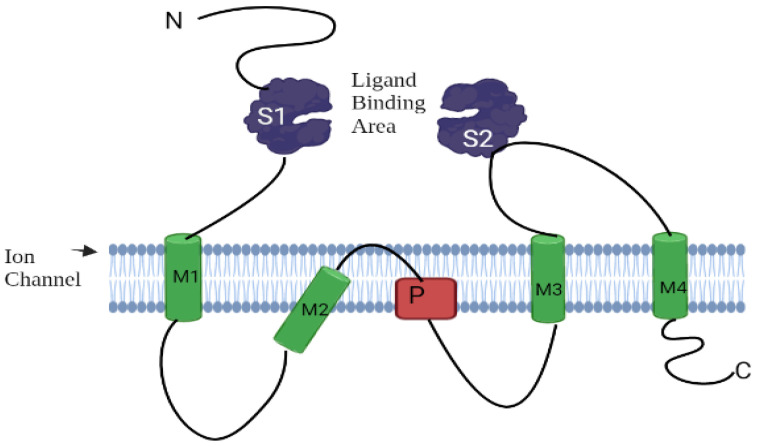
Assembly of a glutamate receptor.

**Figure 4 plants-11-02580-f004:**
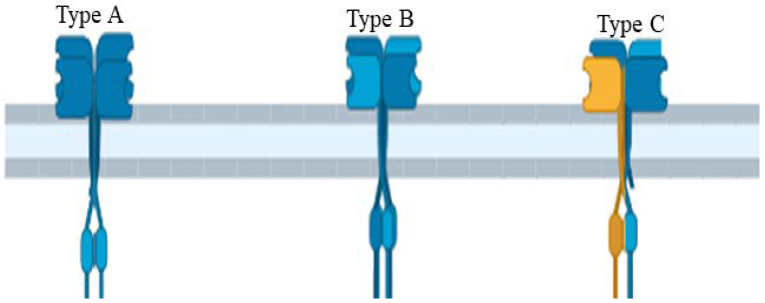
Glutamate receptor channel of *Arabidopsis* showing the wild type and mutant data of hypocotyls.

**Figure 5 plants-11-02580-f005:**
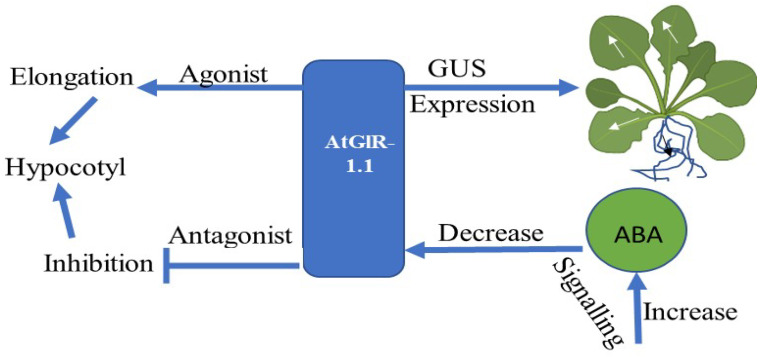
Flowchart showing the expression and function of the AtGluR-1.1 gene.

**Figure 6 plants-11-02580-f006:**
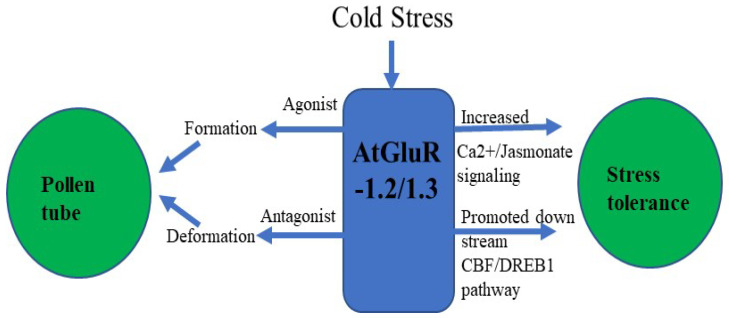
Flowchart of AtGluR-1.2/-1.3.

**Figure 7 plants-11-02580-f007:**
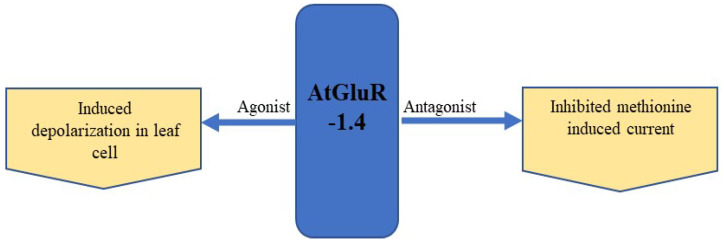
Expression and effect of the agonist and antagonist on AtGluR-1.4.

**Figure 8 plants-11-02580-f008:**
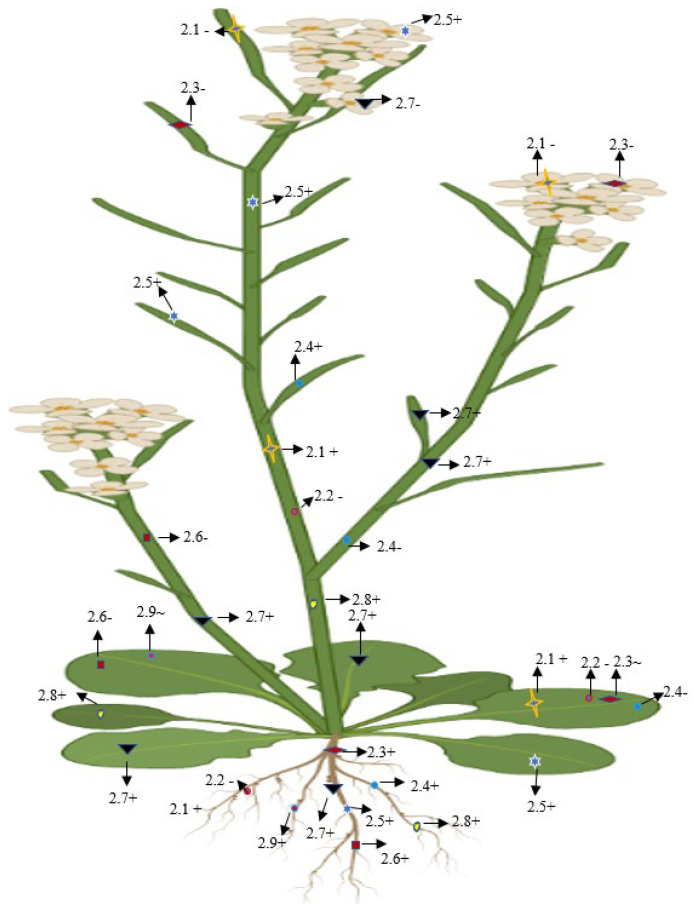
Diagrammatic representation of the presence, absence, and lower expression of Clade II genes in *Arabidopsis.* + (Expression), (no expression), ~ (low expression).

**Figure 9 plants-11-02580-f009:**
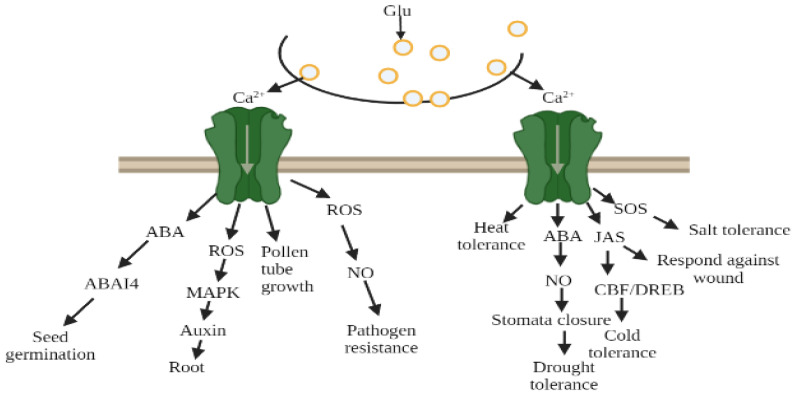
Glutamate receptors mediated by Ca^2+^ ion due to environmental stress, glutamate (Glu), abscisic acid (ABA), reactive oxygen species (ROS), nitric oxide (NO), jasmonate (Jas).

**Figure 10 plants-11-02580-f010:**
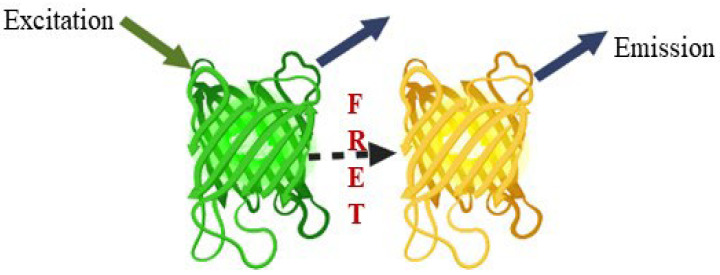
FRET phenomenon, a potential tool.

**Figure 11 plants-11-02580-f011:**
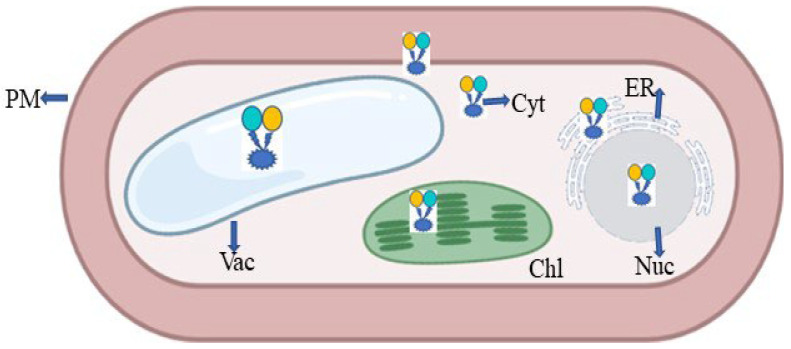
Expression of FRET in the form of nanosensor in a plant cell; PM—plasma membrane; ER—endoplasmic reticulum; Chl—chloroplast; Vac—vacuole; Nuc—nucleus; Cyt—cytoplasm.

**Table 1 plants-11-02580-t001:** List of different agonists, antagonists, or blockers that were used on AtGluRs receptors.

SNo	Agonist	Antagonist/Blocker	Gene Related to Glutamate Receptors	References
1	BMAA		1.1	[14,56]
		DNQX	1.1	[10]
2	D-serine		1.2	[15
		CNQX	1.2	[15]
3	Met, TryTyr, Thr, Leu, Phenylalanine, and Aspargine		1.4	[32,51,52]
		CNQX/AMP/kainiteMK-801, Memantine, and Philanthotoxin	1.4	[10,13,14,15,52,54,55,58,60]
4	Glutamate		2.1	[12]
5	Met		3.1	[61]
6	D-serine, Met, and Glycine		3.2	[15,19,52,61,62]
7	L-glutamate, Glycine, D-Glutamate, GABA, NMDA, Arginine, and Glutamine		3.3, RsGluRs,	[13,15,33,49,50,51,52,54,57]
		DNQX, AP-5	3.3	[55,58,59,60]
8	Glycine, Aspargine, Alanine, L-Serine, D-Serine, and Cystein		3.4	[13,32,33,50,51,52,55]
		L-Alanine, L-Glutamate, and Phenylalanine	3.4	[15,33,49,50]

**Table 2 plants-11-02580-t002:** The expression of Clade III genes in *Arabidopsis*.

S. No.	AtGluRs	Expression Location
1.	3.1	Plasmembrane, guard cell, different cell of seedlings
2.	3.2	All parts of plant
3.	3.3	Leaf and root
4.	3.4	Guard cell, vascular bundles, mesophyll cells, root, stem, seed germination
5.	3.5	Germinated seedNo expression in mature/dry seedcotyledons of the germinating embryo
6.	3.6	Initial stages of root tissue as compared mature root
7.	3.7	Every part of plant

## Data Availability

Not applicable.

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
