# Peer review of "An Insight into Animal Glutamate Receptors Homolog of *Arabidopsis thaliana* and Their Potential Applications—A Review"

_plants, 2022, doi:10.3390/plants11192580_

Round 1
Reviewer 1 Report
This paper by Naz et al. summarized the progress in glutamate receptor proteins in plants. The authors reviewed the current understandings regarding physiological functions, biochemical properties and regulatory mechanisms of glutamate receptor channels in plants. This review article is helpful for the understanding of the mechanism of glutamate receptor in plants. This paper can be accepted for publication in Plants after significant revisions. Especially, figure and the sentence must be improved
1. The authors abbreviated ionotropic glutamate receptors as iGL-rec, though, for a long period of time iGluRs is generally used. Authors also used AtGL-rec for Arabidopsis thaliana glutamate receptor genes. AtGluR is commonly used term. Please change the term to the same as those used generally to prevent confusion.
2. Figure 2 is just listing the AtGLRs and doesn’t provide any important information. Fig. 3 is much important and useful. So Fig. 2 can be deleted.
3. Page 6, line 197-198. Please indicate the name of amino acid which induce gating.
4. Page 6, line 205-206. This is not a text book for first year university students. Agonist and antagonist is a term everybody knows. Explanation of this term is unnecessary.
5. Line 222, “influenced” is unclear. Enhanced or promote is better.
6. Page 7. Authors abbreviated channels “chas”. Authors are making too much original abbreviation. It is not a long word and it is just making hard to read.
7. Page 7. In this pages, authors explaining the character of heteromeric channels, though the term “heteromeric channel“ doesn’t appear in this page. In page 2 and 4, this term is already used so readers know this channel can function in heteromeric form but mentioning in this page well deepens readers understanding.
8. Page 7. Blue color used in Fig.5 type B, the color looks similar and difficult to distinguish.
9. Page 7, line 256. If author used the word “embolism” to explain the situation of clogged pore, it is not proper. When xylem or vascular tissue is stuck, embolism sounds proper. Please consider other expression.
10. Figure 6. Arrow was used for inhibition. Using blunt head arrow is better to represent inhibition. Arrow started from AtGLR1.1 and pointed signalling. It is hard to understand the meaning of this. Does it regulate negatively or positively? Actually, when expression level of AtGLR1.1 decrease, amount of ABA increase in the leaf. Please draw a clear diagram to help readers understanding.
11. Fig 8. Right half of this figure is useless. Just writing down in a sentence is enough.
12. Fig 9. Character is too small to read. Its hard to distinguish “-” and ”~”. Is there any reason for representing the expression pattern of GLR in clade II? If there is not a significant meaning, addition of other clade well help reader.
13. Fig 10. Represent in table is much better for understanding. Is there any reason this figure have to be flowchart? “Flow chat” mistake in typing?
14. Table 1. Explanation about the agonis and antagonist of GLR are explained in page 6 to 7. Arrange table 1 close to page 6 and 7 well help readers understanding.
15. Page 14 line 473. Number of the reference presented in the text and reference number in list doesn’t match. [96, 97] is 95 and 96? Yoshida et al [98] is 97 in the list. Please check the list and renumber.
Minor
Too many mistakes in typing. Did someone check this?
“glu” “Glu”, No “space” where it has to be. No periods after abbreviating Arabidopsis “A thaliana, A. thaliana”.
Line 124, explain “NR” before using abbreviation.
In line 88, authors already wrote fluorescence resonance energy transfer (FRET) , than in line 131, just FRET well be ok.
Line 144, add space after comma.
Line 144, replace arginine and lysine. It better to write in same order of LAOBP.
Line 166, What is glr?
Many abbreviation before reprenting In full name. For example, Line 185, CNQX.
Line 222, “hypocotyl region” Is this ment to be “hypocotyl elongation”?
Line 235 Reference number. Overlap and not in the correct order.
Author Response
Listed Response to Reviewer’s Comments
Manuscript ID: plants-1806751
Manuscript Title: An insight into animal glutamate receptors homolog of Arabidopsis thaliana and their potential applications-a review
Dear Editor,
Thank you very much for your useful comments and suggestions on the structure of our manuscript. All specific amendments to correct error of facts have been made and included in the text at appropriate places. Necessary corrections have been highlighted in text. The detailed corrections are listed below point by point.
Reviewer 1
Comments #
- The authors abbreviated ionotropic glutamate receptors as iGL-rec, though, for a long period of time iGluRs is generally used. Authors also used AtGL-rec for Arabidopsis thaliana glutamate receptor genes. AtGluR is commonly used term. Please change the term to the same as those used generally to prevent confusion.
Ans: We are sincerely thankful for the comments. Term iGluR has been corrected as suggested by reviewer.
- Figure 2 is just listing the AtGLRs and doesn’t provide any important information. Fig. 3 is much important and useful. So Fig. 2 can be deleted.
Ans: Thankyou for suggestion. Figure 2 has been deleted.
- Page 6, line 197-198. Please indicate the name of amino acid which induce gating.
Ans: The suggested corrections have been incorporated
- Page 6, line 205-206. This is not a text book for first year university students. Agonist and antagonist is a term everybody knows. Explanation of this term is unnecessary.
Ans: Unnecessary part has been removed
- Line 222, “influenced” is unclear. Enhanced or promote is better.
Ans: Corrected as suggested
- Page 7. Authors abbreviated channels “chas”. Authors are making too much original abbreviation. It is not a long word and it is just making hard to read.
Ans: We are sincerely thankful for the suggestion, correction has been done.
- Page 7. In this pages, authors explaining the character of heteromeric channels, though the term “heteromeric channel“ doesn’t appear in this page. In page 2 and 4, this term is already used so readers know this channel can function in heteromeric form but mentioning in this page well deepens readers understanding.
Ans: Change has been made as suggested by reviewer
- Page 7. Blue color used in Fig.5 type B, the color looks similar and difficult to distinguish.
Ans: We are thankful for the suggestion due do the inbuilt color in the structure, it is hard to change the color.
- Page 7, line 256. If author used the word “embolism” to explain the situation of clogged pore, it is not proper. When xylem or vascular tissue is stuck, embolism sounds proper. Please consider other expression.
Ans: Thankyou very much for your valuable suggestion. It has been changed accordingly.
- Figure 6. Arrow was used for inhibition. Using blunt head arrow is better to represent inhibition. Arrow started from AtGLR1.1 and pointed signalling. It is hard to understand the meaning of this. Does it regulate negatively or positively? Actually, when expression level of AtGLR1.1 decrease, amount of ABA increase in the leaf. Please draw a clear diagram to help readers understanding.
Ans: Corrected as suggested
- Fig 8. Right half of this figure is useless. Just writing down in a sentence is enough.
Ans: Corrected as suggested
- Fig 9. Character is too small to read. Its hard to distinguish “-” and ”~”. Is there any reason for representing the expression pattern of GLR in clade II? If there is not a significant meaning, addition of other clade well help reader.
Ans: Clear figure has been incorporated
- Fig 10. Represent in table is much better for understanding. Is there any reason this figure have to be flowchart? “Flow chat” mistake in typing?
Ans: Flow chart has been converted in a table form as suggested by reviewer
- Table 1. Explanation about the agonis and antagonist of GLR are explained in page 6 to 7. Arrange table 1 close to page 6 and 7 well help readers understanding.
Ans: Corrected as suggested
- Page 14 line 473. Number of the reference presented in the text and reference number in list doesn’t match. [96, 97] is 95 and 96? Yoshida et al [98] is 97 in the list. Please check the list and renumber.
Ans: Thankyou very much for your suggestion. Corrected as suggested
Minor
glu” “Glu”, No “space” where it has to be. No periods after abbreviating Arabidopsis “A thaliana, A. thaliana”.
Ans : Corrected as suggested
Line 124, explain “NR” before using abbreviation.
Ans: Full form of NR has been incorporated
In line 88, authors already wrote fluorescence resonance energy transfer (FRET) , than in line 131, just FRET well be ok.
Ans: Done
Line 144, add space after comma.
Ans: Done
Line 144, replace arginine and lysine. It better to write in same order of LAOBP.
Ans: Done
Line 166, What is glr?
Ans: It has been written
Many abbreviation before reprenting In full name. For example, Line 185, CNQX.
Ans: Corrected as suggested
Line 222, “hypocotyl region” Is this ment to be “hypocotyl elongation”?
Ans: It has been changed accordingly
Line 235 Reference number. Overlap and not in the correct order.
Ans: Corrected as suggested
Reviewer 2 Report
This manuscript was well-written, the concept and writing flow were perfect, but I still have some comments below:
1.Could the author describe the important of using glutamate receptors homolog in plants especilaay from animal?
2.The author should increase the paragraph of potential application.
3.Some figures were in low resolution.
Author Response
Listed Response to Reviewer’s Comments
Manuscript ID: plants-1806751
Manuscript Title: An insight into animal glutamate receptors homolog of Arabidopsis thaliana and their potential applications-a review
Dear Editor,
Thank you very much for your useful comments and suggestions on the structure of our manuscript. All specific amendments to correct error of facts have been made and included in the text at appropriate places. Necessary corrections have been highlighted in text. The detailed corrections are listed below point by point.
Reviewer 2
1.Could the author describe the important of using glutamate receptors homolog in plants especially from animal?
Ans: Yes, Importance using glutamate receptors homolog has been described
2.The author should increase the paragraph of potential application.
Ans: Potential applications of glutamate receptors has already been discussed in paragraphs from 7 onwards.
3.Some figures were in low resolution.
Ans: Figures have been changed suggested by reviewer and resolution increased up to 300 dpi.